# Revolutionizing MASLD: How Artificial Intelligence Is Shaping the Future of Liver Care

**DOI:** 10.3390/cancers17050722

**Published:** 2025-02-20

**Authors:** Nicola Pugliese, Arianna Bertazzoni, Cesare Hassan, Jörn M. Schattenberg, Alessio Aghemo

**Affiliations:** 1Department of Biomedical Sciences, Humanitas University, 20072 Pieve Emanuele, MI, Italy; nicola.pugliese@humanitas.it (N.P.); arianna.bertazzoni@humanitas.it (A.B.); cesare.hassan@hunimed.eu (C.H.); 2Division of Internal Medicine and Hepatology, Department of Gastroenterology, IRCCS Humanitas Research Hospital, 20089 Rozzano, MI, Italy; 3Endoscopy Unit, Department of Gastroenterology, IRCCS Humanitas Research Hospital, 20089 Rozzano, MI, Italy; 4Department of Internal Medicine II, Saarland University Medical Center, 66421 Homburg, Germany; joern.schattenberg@uks.eu

**Keywords:** fatty liver disease, liver steatosis, deep machine learning, chatbot, metabolic syndrome

## Abstract

In recent years, there has been a progressive and growing interest in artificial intelligence (AI) and its potential applications in the medical field, including hepatology. Given the significant and increasing global prevalence of MASLD and its impact on daily clinical practice, the use of AI in this field could have positive implications for both clinicians and patients. This narrative review aims to summarize the currently available evidence on the potential applications of AI in MASLD, from diagnosis and risk stratification to patient counseling and the development of new treatment options.

## 1. Introduction

Metabolic dysfunction-associated steatotic liver disease (MASLD) is a leading cause of chronic liver disease, with a rapidly increasing incidence worldwide [1]. Since the landmark multi-society Delphi consensus in 2023, MASLD has replaced the previous term Non-alcoholic fatty liver disease *(NAFLD)*. This change addresses several limitations, including ambiguous terminology, inadequate patient stratification, and stigmatizing language [2,3]. MASLD is diagnosed in the presence of steatotic liver disease (SLD) associated with one or more cardio-metabolic risk factors in the absence of harmful alcohol consumption. The spectrum of the disease ranges from isolated metabolic dysfunction-associated steatotic liver disease (MASLD) to metabolic dysfunction-associated steatohepatitis (MASH), progressing to liver fibrosis and ultimately, cirrhosis [1].

In recent decades, the epidemiological profile of liver disease has changed significantly. Effective antiviral therapies such as nucleotide/nucleoside analogues for HBV and direct-acting antivirals (DAAs) for HCV, combined with public health measures such as HBV vaccination and widespread access to DAAs, have led to a decline in viral-induced liver cirrhosis and acute liver failure [4,5,6]. At the same time, the global prevalence of obesity, diabetes, and metabolic syndrome has surged. This, along with improved outcomes and increased life expectancy for cardiovascular patients, has contributed to an increase in diagnoses of MASLD [3,7,8,9]. According to a recent systematic review, the prevalence of MASLD was estimated to be approximately 30% among adults worldwide, with significant regional variations [7].

Interest in the application of artificial intelligence (AI) in medicine is growing rapidly. AI encompasses computer-generated algorithms designed to augment human intelligence in various domains, including drug development, data analysis, integrated patient diagnosis, risk stratification, clinical management, physician education, and patient counseling [10,11,12,13,14]. Despite the immense potential of AI to revolutionize healthcare, its widespread use raises ethical concerns related to data management, patient privacy, potential financial conflicts of interest, etc. [10,15,16].

The term “artificial intelligence” was coined in the 1950s to describe computer programs that simulate human cognition [17]. AI uses predefined sets of rules to solve real-world problems [18,19]. Subfields of AI include generative AI (GenAI), which creates new images, text, or music from existing data, and machine learning (ML), the most widely used form of AI [17]. ML develops predictive algorithms based on a dataset, facilitating decision-making and outcome prediction [20]. AI training methods vary from supervised (guided by human-annotated data) to unsupervised learning (where algorithms identify patterns independently) [21]. Deep learning (DL), a subset of ML, uses multi-layer convolutional neural networks (CNNs) that are capable of analyzing large datasets without human input [20,22].

Given the increasing global burden of MASLD and the growing role of AI in medicine, this review explores the impact of AI and deep learning on the management of MASLD (Figure 1). The following sections discuss AI applications in different areas, ranging from predictive modelling and patient counseling to digital pathology and image-based diagnostics.

## 2. Search Strategy and Inclusion Criteria

A comprehensive search of the literature was conducted to identify relevant studies on the applications of AI in the field of MASLD, with a particular focus on large language models (LLMs), ML, digital pathology, and radiological assessment. A variety of databases were searched, including PubMed, Scopus, and Web of Science, with the results covering publications from 2018–2024.Studies on AI applications in other liver diseases (e.g., autoimmune liver diseases, cirrhosis, and hepatocellular carcinoma, HCC) are included in Table 1 for reference but are not discussed in the main text, in order to maintain the focus on MASLD.

## 3. The Expanding Role of AI in Liver Diseases

In recent years, AI has gained significant attention in hepatology, with applications extending well beyond traditional diagnostics. AI has been used to develop integrated diagnostic tools, improve patient stratification, assess risk, and even explore innovative therapeutic models [13,20,21].

One notable example is the development of predictive algorithms that combine data from electronic medical records (EMRs) with imaging modalities such as ultrasound or computed tomography (CT). These tools have the potential to minimize the need for invasive liver biopsies by providing non-invasive insights into disease etiology and fibrosis staging. Using EMRs and imaging data, deep learning algorithms are able to process large and complex datasets, providing a more personalized and comprehensive understanding of each patient’s risk profile [13,20,32]. The ability of AI to integrate laboratory and imaging data has also opened the door to non-invasive prediction of the hepatic venous pressure gradient (HVPG), as demonstrated in the work of Reiniš et al. [13,33]. Moreover, deep learning-based algorithms have been explored for histopathology applications to augment human assessment and assist in challenging diagnostic scenarios [20,34]. Several recent studies have demonstrated the potential of AI to discriminate between healthy and pathological tissue or to predict outcomes such as early recurrence of HCC after surgery [35,36].

In radiology, AI-based technology has the potential to quantify liver steatosis using imaging modalities such as ultrasound, CT scanning, or magnetic resonance imaging (MRI), helping to predict liver fibrosis and stratify patients at risk of severe HCC disease [13,18,20,21]. AI-based frameworks are also being investigated to improve the performance of transient elastography (e.g., FibroScan). No studies have been published at the time of writing, but the application of AI in this setting has the potential to improve early detection of liver fibrosis and accurately predict patient outcomes [37].

In addition, AI-based chatbots, such as ChatGPT, are emerging as valuable 24/7 tools, providing patients with answers to specific queries, assisting with routine tasks and providing guidance to healthcare professionals. Several studies have validated the accuracy and comprehensibility of these chatbots in various settings, including management of chronic diseases, with growing interest in their application for MASLD patient care [20,23,27,28].

AI is also revolutionizing drug development, not only by assisting with data analysis, but also by facilitating the creation of organ-on-chip (OOC) models for drug testing [19,20].

Moreover, in the context of liver transplantation, AI can optimize patient outcomes by identifying candidates with the highest 1-year mortality risk, predicting post-surgical outcomes, and assessing the risk of complications such as kidney failure [18,21].

## 4. LLMs and MASLD: From Patient Counseling to Histopathological Analysis

One of the emerging applications of AI in healthcare is GenAI, particularly LLMs (Table 1). These conversational tools, such as ChatGPT, are trained on large language datasets and can generate new content by identifying and replicating patterns from their training data [17,38]. ChatGPT, for example, is based on OpenAI’s Generative Pretrained Transformer (GPT) model and has demonstrated its effectiveness in providing answers to a wide range of queries in a variety of healthcare settings, including mental health support and chronic disease management [17,39]. Despite their potential, LLMs also raise concerns, especially around privacy, the adequacy of their training, and the reliability of their output [17,38].

In the context of MASLD, ChatGPT and similar tools could support patient education, provide management guidance to healthcare professionals, and also assist in histopathological interpretation [17,23,26,38].

Several studies have investigated the performance of ChatGPT in responding to MASLD-related queries [23,24,26,39]. A study involving ten key opinion leaders in the field of MASLD evaluated ChatGPT 3.5’s responses to patient queries in English, focusing on accuracy, completeness, and comprehensibility, using three- and six-point Likert scales. The study showed that ChatGPT was able to generate complete and understandable responses, with mean scores of 2.08 ± 0.3 and of 2.87 ± 0.14 on the three-point Likert scale, respectively. However, accuracy was suboptimal, with a mean score of 4.84 ± 0.74 on the six-point Likert scale [24]. A separate study by the same authors aimed to evaluate the chatbot’s performance in Italian by posing the same set of questions to Italian-speaking experts. The results showed a slight deterioration in accuracy (4.57 ± 0.42), while completeness (2.14 ± 0.31) and comprehensibility (2.91 ± 0.07) remained comparable to the previous results [23].

Comparing different LLMs, Zhang et al. found that ChatGPT-4 outperformed other AI-based assistants, achieving an appropriateness rating of 96.7%, while Bard (Google), Llama2 (Meta), and Claude2 (Anthropic) ranged between 80–90% [26]. These findings highlight ChatGPT-4’s superiority in MASLD patient counseling, but also emphasize the need for human oversight, particularly to ensure accuracy.

Moreover, beyond text-based applications, generative AI has the potential to incorporate image-reading capabilities, making it useful in the field of pathology. In the field of MASLD, LLMs can assist pathologists in the diagnosis, grading, and staging of liver steatosis and fibrosis [22]. A study by Zhang et al. demonstrated that ChatGPT-4 achieved 87.5% accuracy in detecting the presence of MASH and assessing the stage of liver fibrosis; these results were comparable to those obtained by a panel of expert pathologists [25]. This example highlights just one of the many evolving applications of AI in MASH pathology, a field that continues to expand in terms of potential applications.

Although the available data are still early, LLMs appear promising for possible future use in clinical practice, both as a counselling tool for patients with MASLD and for possible histological evaluation of liver biopsies.

## 5. AI in Histological Evaluation of MASH: From Biopsy Analysis to Digital Pathology

AI is transforming digital pathology (DP), and has the potential to revolutionize the way biopsy samples are interpreted and diagnosed both in clinical trials and clinical practice [22]. Currently, liver biopsy remains the gold standard for diagnosing MASH and monitoring fibrosis progression [1]. It is also critical in drug development, as histological evaluation is generally required for patient enrollment, assessment of trial endpoints, and evaluation of the treatment’s success [1,22].

Despite its central role, liver biopsy has limitations. The heterogeneity of liver fibrosis and inflammation can lead to under- or over-staging, potentially affecting clinical and research outcomes [22]. Additionally, inter-observer variability poses a significant challenge, with the same sample sometimes receiving different scores from pathologists [12,22,40]. Furthermore, current histological scoring systems lack the precision required for nuanced patient stratification, which can impact the assessment of early therapeutic response [22,40].

In response to these challenges, AI-enabled DP has emerged as a promising tool for achieving more reliable and standardized results [12,22,40].

AI applications in this setting use whole slide imaging (WSI) to scan biopsy samples, which are then analyzed using ML and DL algorithms [40]. These technologies can provide objective assessment by identifying and quantifying histological features such as steatosis and fibrosis. For example, supervised ML models trained with pathologist annotations can accurately identify macro- and microvesicular steatosis or fibrosis [41,42,43]. Additionally, segmentation frameworks have been developed for detailed quantification of vacuoles and other morphological features [44].

Several studies underscore the promise of AI in digital pathology. Vanderbeck et al. developed a supervised machine learning algorithm that achieved an overall accuracy of 89% in identifying steatosis and other anatomical structures (central veins, sinusoids and portal triads). The classifier performed best in identifying macrosteatosis, with an accuracy rate of 95.7%, while its performance for other structures ranged from 61.5 to 91% [41]. Munsterman et al. developed an automated steatosis quantification system that was validated in 61 NAFLD patients and 18 controls, achieving 91.9% accuracy in identifying steatosis [42].

Gawrieh and colleagues developed an automated tool to identify and quantify fibrosis patterns in NAFLD biopsies, using the collagen proportionate area (CPA). Their model showed excellent performance for identifying nodules/cirrhosis and bridging fibrosis (AUROC > 90%), although performance was slightly lower for periportal, pericellular, and portal fibrosis (AUROC 78.6–86.4%) [43].

AI-based digital pathology systems can further improve diagnostic accuracy when integrated with advanced imaging techniques such as second harmonic generation (SHG) microscopy and two-photon excitation fluorescence (TPEF) microscopy. These methods allow precise identification of collagen fibers and liver tissue architecture, supporting automated tools such as the qFibrosis^®^ system [45,46,47,48,49].

For example, Naoumov and colleagues conducted a post hoc analysis of the FLIGHT-FXR trial (NCT02855164), using SHG/TPEF microscopy to quantify fibrosis (qFibrosis^®^), steatosis (qSteatosis^®^), and ballooning (qBallooning^®^) in patients treated with different doses of the antifibrotic drug tropifexor (TXR). The results were then compared with traditional models, such as the NASH Central Research Network (CRN) scoring system and conventional microscopy. Overall, SHG/TPEF microscopy showed greater sensitivity than conventional methods in identifying anti-fibrotic effects in the F3 fibrosis cohort. Specifically, SHG/TPEF microscopy detected anti-fibrotic effects in 50% (11/22) of patients treated with TXR 140 μg and in 83% (15/18) of patients treated with TXR 200 μg, whereas conventional microscopy had a detection rate of 27% (6/22) in the TXR 140 μg group and 17% (3/18) in the TXR 200 μg group. Such increased sensitivity has great potential for the detection of drug-induced histological changes, especially in therapeutic clinical trials [47].

As research in this area continues to evolve, the integration of AI into histological assessment holds great promise for improving diagnostic accuracy, enhancing clinical decision-making, and optimizing patient outcomes in MASLD.

While liver biopsy remains essential for the assessment of fibrosis and inflammation, AI-assisted imaging techniques are increasingly enabling non-invasive approaches to assess liver steatosis and disease progression.

## 6. AI and Radiological Diagnosis of Steatosis: Advancing Non-Invasive Detection

Liver steatosis can be defined histologically or radiologically, with the latter technique being preferred due to its less invasive and more cost-effective nature. Recent technological advancements and the functional limitations of current diagnostic methods have stimulated interest in non-invasive tests (NITs), including the application of AI in radiology [1,18,22,50,51].

The most accessible technique for liver evaluation is conventional B-mode US, which allows rapid assessment of patients with high specificity (97–100%) but relatively low sensitivity (60–64%) and operator-dependent variability [52,53]. Diagnosis and quantification of steatosis is generally based on an assessment of increased echogenicity or peripheral attenuation [54]. Particularly useful is the assessment of the hepato-renal index (HRI), which is defined as the difference in brightness between the liver parenchyma and the renal cortex [52]. The controlled attenuation parameter (CAP), measured during transient elastography, represents another technique for estimating steatosis but one which has limitations due to its lower accuracy [51].

AI-enhanced US techniques have shown promise in improving diagnostic reliability. For example, an AI-based algorithm developed by Santoro et al. was designed to automatically calculate HRI (HRIA). In a study of 134 healthy volunteers, HRIA showed a stronger correlation with MRI-derived proton density fat fraction (MRI-PDFF) results than manually-calculated HRI (HRIM) [52].

Similarly, a study by Cao et al. demonstrated that DL techniques achieved the highest diagnostic ability in differentiating moderate from severe MASLD in the analysis of images from 240 patients, with an AUC of 0.958 [55]. Another study by Kwon et al. introduced an AI-enhanced US coefficient for quantifying liver fat content, with results comparable to MRI [56].

AI applications extend beyond US to CT and MRI imaging [53]. In CT scans, liver fat content is measured by the attenuation of signal, measured on the Hounsfield unit (HU) scale. As triglycerides absorb less X-rays than normal hepatocytes, a lower HU value corresponds to higher fat content [57]. Typically, 64 HU indicates no steatosis on histology, while 42 HU corresponds to moderate steatosis. The specificity and sensitivity of CT vary with severity, reaching 95% and 75% for severe steatosis [53]. Quantification of liver fat content can be performed directly using liver HU sampling or indirectly using the liver–spleen HU difference (in contrast-enhanced studies) [53]. AI algorithms are being developed to improve the accuracy of fat quantification by addressing challenges such as uneven fat distribution across the liver parenchyma.

In MRI, fat quantification is primarily achieved using PDFF, which calculates the proportion of mobile protons attributed to liver fat [58]. Advanced techniques such as magnetic resonance spectroscopy (MRS) and chemical shift-encoded (CSE) MRI allow precise separation and quantification of signals from water and fat. These methods have shown strong correlation with histopathological findings [53,58,59,60]

AI-driven approaches further improve fat quantification by automating analysis in specific regions of interest (ROIs) [53]. However, due to the heterogeneous distribution of fat in the liver parenchyma, further studies are required to standardize ROI selection and integrate data from multiple regions to ensure reliable results.

Advances in AI-driven imaging are complemented by machine learning models capable of integrating clinical, biochemical, and imaging data to refine risk stratification and prognosis in MASLD.

## 7. Machine Learning in MASLD: Predictive Modeling and Risk Stratification

Given the increasing prevalence of MASLD, it is critical to identify patients at risk of advanced chronic liver disease or HCC to enable appropriate follow-up and timely intervention that can improve prognosis [12]. One promising approach is to use EMRs, which contain vast amounts of categorized patient data, to develop predictive models that can identify high-risk individuals [12].

As an example, the study by Fialoke et al. used one of the largest US EMRs, Optum^©^ Analytics, to develop ML models to predict MASH in MASLD patients, and the most accurate model was selected. Specifically, the Optum^©^ EMR was used to develop class-balanced patient cohorts with or without steatotic liver disease. Secondly, supervised classification of patients was performed with different MLMs according to demographic characteristics, comorbidities, and temporal mean values from laboratory tests (AST, ALT, platelet count). Finally, the best performing models were applied to another cohort of patients and evaluated for accuracy. Overall, four models were selected as the best performers: logistic regression, decision tree, random forest, and XGBoost. These supervised learning models outperformed existing NITs, achieving AUC values ranging from 0.83 to 0.88. Of these, the XGBoost model showed the highest accuracy for MASH prediction [61].

Another notable ML application is the NASHMap^©^ model, designed to predict MASLD and MASH in high-risk patients. The model includes 14 key parameters and was trained on two large patient datasets: the Optum^©^ EMR and the National Institute of Diabetes and Digestive and Kidney Diseases (NIDDK) registry. Overall, the model demonstrated a satisfactory performance in predicting MASH in both databases (AUC of 0.82 in the NIDDK registry and 0.76 in the Optum^©^ EMR). To adapt to the variable availability of data in clinical practice, a five-feature algorithm was developed, which showed slightly lower but still satisfactory performance [62].

In addition to identifying MASH, ML models have also shown promise in predicting the development of HCC, a critical step in improving patient prognosis through early intervention. A recent study by Sarkar et al. developed ML models based on EMR datasets from the University of California, Davis (UC Davis) and the UC, San Francisco (UCSF). These models achieved an impressive AUC of 0.97 for HCC prediction. Among the identified HCC predictors, liver fibrosis as determined by the FIB-4 score emerged as the strongest determinant [63].

## 8. AI and Drug Development for MASLD

Current management of MASLD is complex and relies primarily on non-pharmacological interventions, including appropriate diet and physical activity, with the goal of achieving a 5–10% weight loss [1]. In some cases, pharmacological, endoscopic, or surgical weight loss options may also be considered [1].

However, effective pharmacological therapies that target not only hepatic fat accumulation but also inflammation and fibrosis remain limited. The only approved drug for MASH, resmetirom, is limited to selected patient populations, highlighting the need for novel therapeutic strategies [1]. To address this challenge, AI is playing an increasingly crucial role in optimizing drug discovery [20,64].

AI can accelerate drug discovery by identifying new therapeutic targets and optimizing screening methods. For example, Xia et al. used AI to overcome limitations in the development of FXR agonists. In a previous study, structure-based virtual screening (SBVS), which is a widely used computational approach to drug discovery, failed to effectively identify FXR agonists, probably due to limitations in accounting for protein flexibility. To improve accuracy, Xia et al. developed a human FXR (hFXR)-specific learning model based on pose filters from 24 agonist-bound hFXR crystal structures and integrated it with traditional SBVS approaches (FRED docking and Chemgauss 4 scoring function). This approach ultimately identified a novel potential therapeutic strategy for further drug development [65].

Another cutting-edge application of AI in drug development involves organ-on-chips (OOC) technology. These three-dimensional platforms replicate the physiological environment of human organs, providing a powerful tool for preclinical drug testing. Hepatological OOC models are being refined to better mimic the three-dimensional liver architecture, including cellular organization and physiological fluid flow. Early studies have demonstrated promising results in liver tissue modeling [66]. In this context, MLM and DL can enhance drug efficacy evaluation and the detection of early pathological changes at the cellular level [19]. A notable example was reported by Chu et al. who developed an AI-driven automated system for drug testing using microfluidic monitoring. Their system employed a convolutional neural network to analyze microcapsule production in real time, effectively classifying outcomes as “good” or “bad” and integrating this information into an adaptive valving system. This AI-based approach reduced human error, improved efficiency, and lowered production costs, offering a scalable solution for high-throughput drug screening [67].

The integration of AI-driven approaches in MASLD drug development has the potential to revolutionize therapeutic strategies, accelerating target identification, preclinical testing, and clinical translation. As AI models become more sophisticated, further validation and regulatory adaptation will be essential to ensure their safe and effective implementation in clinical practice.

## 9. Limitations and Ethical Concerns in AI-Driven MASLD Management

While AI has generated significant enthusiasm within the scientific community, its integration into healthcare is not without challenges and concerns. A key issue is the potential for bias in AI models, particularly when they are developed using datasets from single centers with variable data quality. These biases can lead to inaccurate predictions and reduced generalizability. To address this, AI models should be trained on large, multi-center datasets representing diverse populations, to ensure robustness and fairness. Furthermore, extensive training on well-characterized patient cohorts is essential before using these models in clinical practice [68].

Another notable limitation is the risk of AI ‘hallucinations’, where LLMs generate incorrect or misleading outputs that appear plausible [69]. Such errors, if misinterpreted as factual, can have serious consequences for patient care, underscoring the critical need for careful human oversight in clinical decision-making. Ethical considerations should also be kept in mind. Indeed, AI may deprive patients of autonomy (e.g., in the case of over-reliance on LLM-based recommendations), or it could potentially be subject to manipulation or misinformation (thus undermining the principle of non-maleficence). Moreover, it is essential to ensure that the benefits to patients clearly and transparently exceed the commercial obligations of companies developing AI models (beneficence principle) and that access to AI tools is equitable for all (justice principle) [14].

Lastly, personal patient data should be managed and treated appropriately, according to local regulations, with transparent indications on how patient-level data are handled and stored, to avoid jeopardizing data confidentiality [14,70,71,72].

## 10. Conclusions

The global increase in MASLD is a pressing challenge for healthcare systems worldwide, requiring the development of innovative diagnostic and management strategies. Early detection and timely intervention are essential to strategies for preventing the progression of MASLD and mitigating its associated complications, including HCC.

AI offers immense potential to improve the management of MASLD patients. Its integration into diagnostic workflows—from non-invasive imaging techniques to predictive algorithms—can improve patient stratification, facilitate personalized treatment plans, and identify individuals at higher risk of serious outcomes who require closer monitoring.

However, to unlock the full potential of AI, future research must prioritize the validation of AI-developed models on large, diverse patient databases. This will ensure robust performance, uncover potential limitations, and establish the reliability of these tools before integration into clinical practice.

Finally, while the integration of AI into MASLD management has the potential to revolutionize patient care, it should complement and not replace human expertise. Judicious use of AI alongside clinical oversight is essential for maintaining the holistic and ethical dimensions of patient care. Ongoing research, combined with a commitment to ethical standards, will be essential to fully realize the transformative potential of AI in MASLD. 

## Figures and Tables

**Figure 1 cancers-17-00722-f001:**
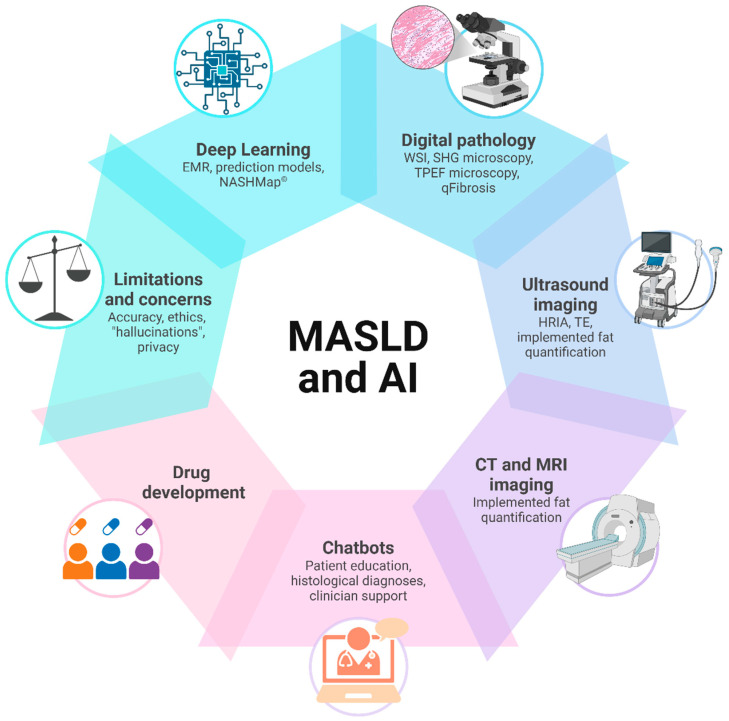
Overview of possible applications of artificial intelligence (AI) in the field of metabolic dysfunction-associated steatotic liver disease (MASLD). EMRs: electronic medical records; HRIA: automated hepato-renal index; SHG: second harmonic generation; TE: transient elastography; TPEF: two-photon excitation fluorescence; WSI: whole-slide imaging. Realized with BioRender https://BioRender.com/j10b220.

**Table 1 cancers-17-00722-t001:** Studies published so far evaluating AI-based chatbot performance in liver diseases. LLM: large language model; EQIP: Ensuring Quality Information for Patients; HCC: hepatocellular carcinoma; AILD: autoimmune liver diseases; AIH: autoimmune hepatitis. * “appropriate” when they were free from errors and “inappropriate” when they contained potential factual errors. ^§^ “accurate” score 1, if all information is true and relevant; “inadequate” score 0, if all information is true, but does not fully answer the question or provides irrelevant information; “inaccurate” score-1, if any information is false.

Authors	Year of Publication	Topic	Chatbot/LLM	Scoring System	Items	Results	Notes
**MASLD**							
Pugliese et al. [23]	2024	LLM as a couseling tool for MASLD patients	ChatGPT-3.5 (OpenAI)	Likert scale	Accuracy (six-point scale) Completeness (three-point scale) Comprehensibility (three-point scale)	Median scores Accuracy 4.57 ± 0.42 Completeness 2.14 ± 0.31 Comprehensibility 2.91 ± 0.07	Evaluation of responses by 13 experts
Pugliese et al. [24]	2024	LLM as a couseling tool for MASLD patients	ChatGPT-3.5 (OpenAI)	Likert scale	Accuracy (six-point scale) Completeness (three-point scale) Comprehensibility (three-point scale)	Median scores Accuracy 4.84 ± 0.74 Completeness 2.08 ± 0.51 Comprehensibility 2.86 ± 0.14	Evaluation of responses by three experts
Zhang et al. [25]	2024	LLM for histological grading of MASH	ChatGPT-4 (OpenAI)	-	Identification of MASH and fibrosis	ChatGPT-4: 87.5% accuracy	Evaluation of responses by two experts
			Bard (Google)			Bard: 38.3% accuracy	
Zhang et al. [26]	2023	LLM as a couseling tool for MASLD patients	ChatGPT-3.5 (OpenAI)	-	Appropriateness *	ChatGPT-3.5: 80% appropriateness	Evaluation of responses by three experts
			ChatGPT-4 (OpenAI)			ChatGPT-4: 96.7% appropriateness	
			Bard (Google)			Bard: 90% appropriateness	
			Llama2 (Meta)			Llama 2: 90% appropriateness	
			Claude2 (Anthropic)			Claude2: 80% appropriateness	
Other liver diseases							
Daza et al. [27]	2024	LLM as a couseling tool for AILD patients	ChatGPT-3.5 (OpenAI)	Likert scale	Quality of answers	ChatGPT-3.5: mean score 7.17 (SD = 1.89)	Evaluation of responses by 10 experts
			Claude (Anthropic)			Claude: mean score 7.37 (SD = 1.91)	
			Copilot (Microsoft)			Copilot: mean score 6.63 (SD = 2.10)	
			Bard (Google)			Bard: mean score 6.52 (SD = 2.27)	
Colapietro et al. [28]	2024	LLM as a couseling tool for AIH patients	ChatGPT-4	Likert scale	Accuracy (6 points scale) Safety (5 points scale) Completeness (3 points scale) Comprehensibility (3 points scale)	Median scores Accuracy 5 (IQR 4–6) Safety 4 (IQR 4–5) Completeness 2 (2–2) Comprehensibility 3 (2–3)	Evaluation of responses by 11 experts
Yeo et al. [29]	2024	Evaluation of LLM responses to common questions on cirrhosis and HCC	ChatGPT-3.5 (OpenAI)	Four grades: comprehensive, correct but inadequate, mixed correct and incorrect/outdated data, completely incorrect	-	Cirrhosis: Comprehensive: 49.45% Correct but inadequate: 30.77% Mixed with correct and incorrect/outdated data: 19.78% Completely incorrect: 0% HCC: Comprehensive: 41.1% Correct but inadequate: 32.87% Mixed correct and incorrect/outdated data: 19.18% Completely incorrect: 6.85%	Evaluation of responses by three experts
Cao et al. [30]	2024	Evaluation of LLM responses to common questions on HCC	ChatGPT-3.5 (OpenAI)	Three grades: accurate, inadequate, inaccurate ^§^ Flesch Reading Ease Score and Flesch-Kincaid Grade Level for readability	Accuracy Reliability Readability	ChatGPT-3.5: 45% accuracy; 30% accuracy and reliability	Evaluation of responses by six experts
			Gemini (Google)			Gemini: 60% accuracy; 40% accuracy and reliability	
			Bing (Microsoft)			Bing: 30% accuracy, 15% accuracy and reliability	
Walker et al. [31]	2023	Evaluation of LLM responses to common questions on cirrhosis, HCC, pancreatic disorders	ChatGPT-4 (OpenAI)	Modified EQIP tool (max score: 36 points)	Three sections: Content (18 points) Identification (6 points) Structure data (12 points)	Content: 10 (IQR 9.5–12.5) Identification: 1 (IQR 1–1) Structure data: 4 (IQR 4–5)	Evaluation of responses by two experts

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
