# Peer review of "Revolutionizing MASLD: How Artificial Intelligence Is Shaping the Future of Liver Care"

_cancers, 2025, doi:10.3390/cancers17050722_

Round 1
Reviewer 1 Report
Comments and Suggestions for Authors
Authors aimed to provide a comprehensive overview of the applications of AI in hepatology, with a focus on MASLD, highlighting both its
transformative potential and its inherent limitations. This is an interesting paper.
There are several minor points to be addressed.
1) This review primarily focus on the large-language model based application. So, how to integrate the raw image into clinical practice using deep learning
should be addressed.
2) Table 1 should include more information about performance. Were they developed for diagnosis, transfer to special clinic or prognostication?
Author Response
Comment 1: This review primarily focus on the large-language model based application. So, how to integrate the raw image into clinical practice using deep learning
Response 1: We renamed the paragraph ‘Chatbots and MASLD’ as ‘LLMs and MASLD’. In this paragraph, the potential clinical applications of LLMs in MASLD are explained on the basis of currently available scientific evidence. We have also included a final sentence summarising future perspectives.
Comment 2: Table 1 should include more information about performance. Were they developed for diagnosis, transfer to special clinic or prognostication?
Thank you for this important comment. We have modified the table to make it clearer.
Reviewer 2 Report
Comments and Suggestions for Authors
I have examined your study titled "Revolutionizing MASLD: How Artificial Intelligence is Shaping the Future of Liver Care" in detail. I have listed the points I found lacking in the study. The sentence "For AI to be fully integrated into clinical practice, it must be complement rather than replace human expertise." in the abstract section does not fully fit the meaning. There is no such claim. If there is, it can be discussed in the discussion section. The last paragraph of the introduction section should be finished with a paragraph that includes the organization of the article. A paragraph that highlights the innovative aspects of the article and its contributions to the literature should be added before this. The remaining sections of the article are very disconnected from each other. The other main headings are independent of each other and contain general information. Some studies are examined in Table 1. However, considering that the study is a review article, these articles should be discussed under a title such as related work in advance. Some of the most important deficiencies are that the article does not include important points such as which studies were included in the article, which years were scanned, and which databases the articles were retrieved from. It is possible to expand such deficiencies.
Comments on the Quality of English LanguageSpelling and grammatical errors should be reviewed.
Author Response
Comment 1: "For AI to be fully integrated into clinical practice, it must be complement rather than replace human expertise." in the abstract section does not fully fit the meaning. There is no such claim. If there is, it can be discussed in the discussion section".
Response 1: We appreciate the reviewer’s feedback. As suggested, we have removed this sentence from the abstract to ensure consistency with the main text.
Comment 2: The last paragraph of the introduction section should be finished with a paragraph that includes the organization of the article. A paragraph that highlights the innovative aspects of the article and its contributions to the literature should be added before this
Response 2: We appreciate the reviewer’s valuable suggestion. As recommended, we have added a paragraph highlighting the innovative aspects of our work and its contributions to the literature.
Comment 3: The remaining sections of the article are very disconnected from each other. The other main headings are independent of each other and contain general information.
Response 3: We appreciate the reviewer’s constructive feedback. To enhance coherence, we have: 1) Added transition sentences at the end of each section to improve logical flow, 2) Refined section titles to reflect their interconnections and ensure continuity.
Comment 4: Some studies are examined in Table 1. However, considering that the study is a review article, these articles should be discussed under a title such as related work in advance.
Response 4: We appreciate the reviewer’s suggestion and have revised the manuscript accordingly. The section on LLMs in MASLD has been expanded to better align with the data presented in Table 1. We have incorporated additional details regarding the performance of ChatGPT and other LLMs in MASLD-related applications, particularly in patient counseling and histopathology assessment. While the table includes data from studies evaluating LLMs in other liver diseases, we have opted to keep these results in the table without integrating them into the main text, ensuring a focused discussion on MASLD while maintaining transparency about broader research efforts in the field. These modifications provide a more structured and relevant analysis of LLM applications in MASLD.
Comment 5: Some of the most important deficiencies are that the article does not include important points such as which studies were included in the article, which years were scanned, and which databases the articles were retrieved from. It is possible to expand such deficiencies.
Response 5: We appreciate the reviewer’s feedback. We have added a Methods section specifying the study selection criteria, the timeframe of the literature search, and the databases used to enhance transparency in our review process.
Comment 6: Spelling and grammatical errors should be reviewed.
Response 6: We have extensively reviewed the entire manuscript.
Reviewer 3 Report
Comments and Suggestions for Authors
The submitted review article is focused on benefits and limits of AI application in MASLD, formerly known as NAFLD. Having in mind, that MASLD is a public health issue, being not only a major cause of liver-related morbidity and mortality, but also an independent risk factor for the development of noncommunicable diseases, the topic of this article is important, especially regarding the fact that nowadays, there is no definite treatment for MASLD. Based on the fact that this is just a narrative review paper some other issues need to be addressed.
-Keywords: Please avoid using already present words in the title. Insert fatty liver disease, deep machine learning…
-Line 38: please check the abbreviation MASL?D?
-In introduction, please insert few lines regarding the current pharmacological treatment approach in MASLD treatment
-In introduction, please insert few lines regarding the potential mechanism of MASLD and targeted receptors (See recently published paper doi: 10.3390/cimb47010051) as well as application of in silico techniques in development of new drugs for MASLD (See recently published paper https://doi.org/10.3390/nu16081226 https://doi.org/10.3390/cimb47010051 https://doi.org/10.3390/cells13221827 )
-It will be useful to add section AI in drug development. The part related to application of AI in drug development is missing with few major examples and facts
-Line 240: Please check author guidelines for headings
Author Response
Comment 1: Keywords: Please avoid using already present words in the title. Insert fatty liver disease, deep machine learning
Response 1: We have updated the keywords
Comment 2: Line 38: please check the abbreviation MASL?D?
Checked and corrected
Comment 3: In introduction, please insert few lines regarding the current pharmacological treatment approach in MASLD treatment. In introduction, please insert few lines regarding the potential mechanism of MASLD and targeted receptors (See recently published paper doi: 10.3390/cimb47010051) as well as application of in silico techniques in development of new drugs for MASLD (See recently published paper https://doi.org/10.3390/nu16081226 https://doi.org/10.3390/cimb47010051 https://doi.org/10.3390/cells13221827 )
Response 3: We have included a new paragraph on the usefulness of AI in drug development for MASLD. In the initial part of the paragraph, we mentioned the current therapeutic approach to this disease. We have also assessed with interest the reviewer's suggested bibliography and included two of the suggested papers in our bibliography.
Comment 4: It will be useful to add section AI in drug development. The part related to application of AI in drug development is missing with few major examples and facts.
Response 4: We thank the reviewer for this comment. We have added a new section focused on drug developement.
Round 2
Reviewer 2 Report
Comments and Suggestions for Authors
You have made a severe revision. Congratulations.